# Threatened Aquatic Plants of the Southern Tigris-Euphrates Basin: Status, Threats, and Conservation Priorities

**DOI:** 10.3390/plants14131914

**Published:** 2025-06-22

**Authors:** Murtada Naser, Amaal Yasser, Jonas Schoelynck, Franz Essl

**Affiliations:** 1School of Environment and Science, Griffith University, 170 Kessels Road, Nathan, QLD 4111, Australia; a.ghaziyasser@gmail.com; 2Division of BioInvasions, Global Change and Macroecology, Department of Botany and Biodiversity Research, University of Vienna, Rennweg 14, 1030 Vienna, Austria; franz.essl@univie.ac.at; 3Department of Biology, ECOSPHERE Research Group, University of Antwerp, Universiteitsplein 1C, 2610 Wilrijk, Belgium; jonas.schoelynck@uantwerpen.be

**Keywords:** aquatic plants, biodiversity, conservation, Mesopotamian Marshes, native species, Tigris-Euphrates basin, wetland restoration

## Abstract

The Tigris-Euphrates basin hosts a diverse assemblage of native aquatic plants vital to the region’s ecological and cultural heritage. However, decades of hydrological alterations, pollution, salinity intrusion, habitat destruction, and climate change have caused significant declines in aquatic plant species diversity. This review compiles historical and contemporary information on key native aquatic plant species, assesses their current conservation status, identifies major threats, and provides recommendations for their protection. Sensitive submerged and floating species, including *Vallisneria spiralis*, *Najas marina*, and *Potamogeton* spp., have been particularly affected, with many now being rare or locally extinct. Although restoration efforts in the Mesopotamian Marshes have partially restored some wetlands, aquatic plant conservation remains largely overlooked. We propose targeted recovery plans, integration of aquatic plants into wetland management, enhancement of water quality measures, and increased cross-border hydrological cooperation. Protecting native aquatic flora is essential for maintaining the ecological integrity and resilience of the Tigris-Euphrates basin.

## 1. Introduction

Aquatic plants are fundamental components of freshwater ecosystems, providing essential ecological functions such as oxygen production, sediment stabilization, nutrient cycling, and enhancing physical habitat complexity for a wide array of organisms. In river basins and wetlands, they contribute significantly to ecosystem resilience, particularly under fluctuating hydrological conditions [1,2,3].

The Tigris-Euphrates basin spans parts of Iraq, Syria, Turkey, and Iran. It hosts a unique assemblage of aquatic and semi-aquatic plants adapted to diverse freshwater habitats, including rivers, marshes, lakes, and irrigation canals. Historically, this basin harbored a rich and diverse aquatic vegetation, especially in the Mesopotamian Marshes. These were once the largest wetlands in in the Middle East and among the largest in Asia, particularly in Western Asia and a cradle of human civilization [4,5].

However, in the past decades, aquatic vegetation in the Tigris-Euphrates basin has faced unprecedented threats. Hydrological changes on a large scale, pollution, draining wetlands, salinization, and climate change have devastated aquatic habitats, and have caused substantial declines of native plants [6,7,8,9]. Remote-sensing-based analyses show the extent of loss of vegetation across major marshes, amounting to declines of more than 80% in Al-Hammar (87%), Central (99%), and Al-Huwaiza (84%) marshes over the period from 1982 to 2017 [10]. While aquatic plants are ecologically important, they have received far less attention from conservationists than vertebrates, i.e., fish, birds and mammals [5]. This review focuses on a subset of native aquatic plant species including submerged, floating, and emergent (marsh) macrophytes from the southern Tigris-Euphrates basin that meet one or more of the following criteria: (1) species historically documented as abundant in key wetlands (e.g., Al-Hammar, Central, and Al-Huwaiza marshes); (2) species that have experienced documented or inferred population declines based on recent floristic surveys; (3) species with significant ecological roles in aquatic ecosystems (e.g., oxygenation, habitat structuring); and (4) species highlighted in previous assessments of regional wetland biodiversity or considered potentially threatened by hydrological, salinity, or pollution stress. Widespread or resilient species such as *Phragmites australis*, which tolerate habitat disturbance, were excluded from this synthesis.

## 2. Historical Overview of Aquatic Flora in the Tigris-Euphrates Basin

The Tigris-Euphrates Basin, which spans across Iraq, Syria, Turkey, and Iran, comprises a complex network of rivers, tributaries, floodplains, and wetlands. Within this larger hydrological system, the Mesopotamian Marshes is located in the southern part of Iraq and represent one of the most ecologically significant and unique wetland systems in the region (Figure 1). Historically, the Mesopotamian Marshes were the largest wetland complex in the Middle East and among the largest in Asia. These marshes are fed by the lower reaches of the Tigris and Euphrates rivers and are composed of three main wetland units: Al-Hammar, Central, and Al-Huwaiza Marshes.

While this review references broader hydrological and ecological processes in the Tigris-Euphrates basin, its primary focus is on the macrophyte biodiversity of the Mesopotamian Marshes, due to their exceptional biological richness and the severity of ecological degradation they have experienced.

Ancient Mesopotamian civilizations flourished alongside vast marshes, lakes, and river systems sustained by the annual flooding cycles of the Tigris and Euphrates rivers [11,12,13,14,15]. These dynamic freshwater environments harbored a diverse array of aquatic and semi-aquatic plant species, adapted to a wide range of hydrological and salinity conditions. Early ethnographic and ecological accounts noted the extensive marsh vegetation and the close interdependence between human activities and aquatic ecosystems in southern Iraq [16].

Early botanical surveys, particularly those conducted in the 20th century, documented a rich diversity of submerged, floating, and emergent aquatic plants in the Mesopotamian Marshes of southern Iraq. A detailed ecological study conducted between 1972 and 1975 recorded 371 plant species across the wetlands, of which approximately 40% were wetland-dependent (obligate)or wetland-tolerant (facultative) species [17]. These baseline findings underscore the historical richness of aquatic vegetation in the region. Subsequent assessments, including those by Nature Iraq and Iraq’s Ministry of Environment, confirmed the continued presence of wetland macrophyte communities in the southern marshes despite increasing environmental pressures from agricultural runoff, upstream water diversion, and oil-related activities [18]. Together, these studies highlight the ecological importance and resilience of macrophyte assemblages in Iraq’s marsh ecosystems.

Submerged species such as *Vallisneria spiralis*, *Potamogeton crispus*, and *Najas armata* were abundant in shallow to moderately deep permanent waters (typically 0.5–2 m depth) where light penetration was adequatewhile floating plants like *Nymphoides peltata* and emergent species including *Phragmites australis*, *Typha domingensis*, and *Scirpus litoralis* dominated shallow and seasonally flooded habitats [4,17]. Environmental factors such as water depth, salinity gradients, and soil characteristics strongly influenced the spatial distribution of aquatic plant communities [17].

**Figure 1 plants-14-01914-f001:**
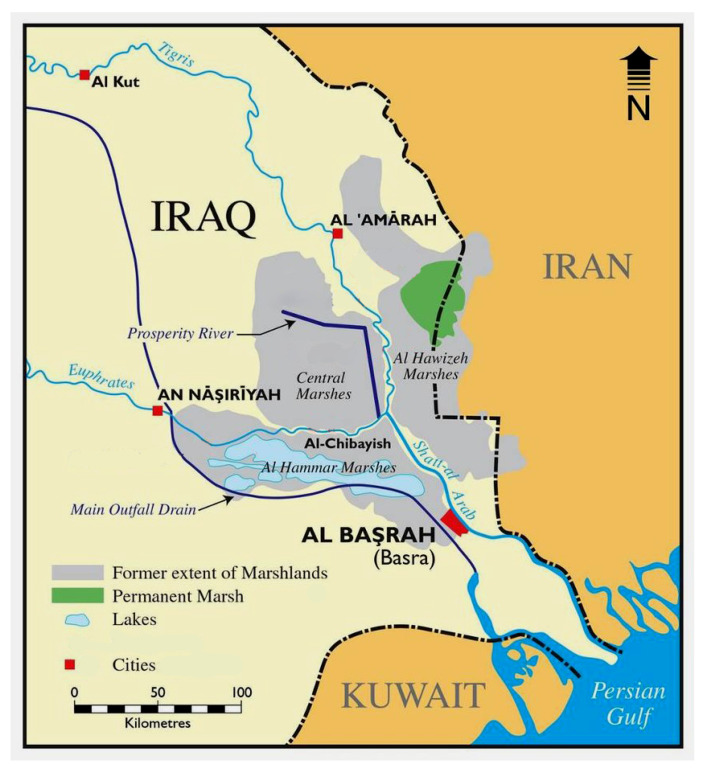
Map of southern Iraq showing (i) the former extent of the Mesopotamian Marshlands, and (ii) the current remaining permanent marsh straddling the Iran-Iraq border (Al-Hawizeh marshes) sourced from [19].

Throughout much of the historical period, traditional practices such as seasonal grazing and fishing maintained a delicate balance between human activity and ecological integrity. However, during the latter half of the 20th century, especially in the 1980s and 1990s, extensive drainage and changes in river flow caused severe environmental stress, resulting in widespread degradation of aquatic habitats [4,13].

Recent surveys in the East Hammar Marshes following partial restoration efforts have provided valuable floristic data on aquatic vegetation. Studies documented a rich assemblage of emergent, submerged, and floating species, including *Phragmites australis*, *Typha domingensis*, *Schoenoplectus litoralis*, *Ceratophyllum demersum*, *Myriophyllum spicatum*, *Najas marina*, *Potamogeton crispus*, *Potamogeton lucens*, *Vallisneria spiralis*, *Salvinia natans*, and *Lemna minor* [20]. However, despite the presence of several submerged species, their abundance was generally low and confined to small, isolated patches, which indicates that environmental conditions remain insufficient for full recolonization and ecosystem recovery. This pattern reflects incomplete ecological recovery and persistent environmental stresses in the marshes, such as turbidity, salinity, and altered hydrology.

Complementary surveys during the 2008 Habitat Mapping and Monitoring Project further documented the presence of key aquatic macrophytes across the Central Marshes and Abu Zirig, including *Phragmites australis*, *Typha domingensis*, *Schoenoplectus litoralis*, *Potamogeton lucens*, and *Ceratophyllum demersum*, providing additional post-restoration baseline data [21].

Surveys conducted by [22] during 2006–2007 documented 44 aquatic macrophyte species in the Huwaiza, Chebaish, and East Hammar Marshes, revealing partial but uneven recovery of submerged, emergent, and floating plants after restoration efforts.

Dominant aquatic plants historically included emergent species like *Phragmites australis*, *Typha domingensis*, and *Schoenoplectus litoralis*, as well as submerged species such as *Vallisneria* spp., *Ceratophyllum demersum*, and *Potamogeton lucens*, which collectively supported rich faunal communities [23].

Aquatic macrophytes play critical roles in wetland ecosystems such as regulating trophic interactions, and enhancing water quality [24]. The decline of submerged plants such as *Vallisneria spiralis*, *Najas marina*, and *Potamogeton* spp. in the Tigris-Euphrates basin therefore reflects not only local habitat degradation, but also a loss of essential ecosystem functions [25,26].

Of particular note are the extensive drainage of the Mesopotamian Marshes as well as human-related activities in the early to mid-1990s, which caused a near-total collapse of the native aquatic plant communities in extensive areas. While partial restoration after 2003 allowed some recovery of some of the plant communities, many sensitive species did not recover so that the overall aquatic plant diversity is still greatly affected by human actions, and has a long way to go before it returns to its previous diversity [27,28,29,30] (Figure 2).

Recent surveys conducted in southern Iraq, particularly in the Basrah region following marshland restoration, have confirmed a partial recovery of emergent macrophyte communities such as *Phragmites australis* and *Typha domingensis* [31]. However, submerged species including *Vallisneria spiralis*, *Najas marina*, and *Potamogeton* spp. remain rare or patchily distributed, reflecting ongoing hydrological and ecological stresses.

## 3. List of Threatened Aquatic Plant Species

The species listed in Table 1 represent a targeted subset of native aquatic macrophytes from the Mesopotamian Marshes that fulfill one or more of the following criteria: (1) historically dominant or commonly recorded species in floristic surveys from the 20th century e.g., [32]; (2) species that have shown clear evidence of population decline or local extinction in post-restoration monitoring e.g., [22]; (3) species with important ecological functions (e.g., water oxygenation, habitat structuring); and (4) species sensitive to key stressors, such as salinity, turbidity, or habitat desiccation. Species that are widespread, highly resilient, or lacked sufficient data on population trends were excluded from this table to maintain focus on the most conservation-relevant taxa (Table 1).

## 4. Threats

Native aquatic plants of the southern Tigris-Euphrates basin face multiple, interlinked threats that have intensified over recent decades. These pressures have led to declines in species richness, habitat fragmentation, and in some cases, local extinctions. Understanding the nature and severity of these threats is crucial for developing effective conservation strategies [36,37,38,39].

### 4.1. Hydrological Alterations

Large-scale hydrological changes represent one of the most critical threats to aquatic plants in the study region. The dams, barrages and irrigation canals built in Turkey, Syria, and Iraq since the mid-20th century have greatly reduced volumes and seasonality of downstream river flows, and in combination with climate change, have significantly altered the water balance leading to the disruption of natural flood pulses and long-term water scarcity [40]. In particular, the disruption of natural flood pulses due to dam construction and irrigation withdrawals has led to the gradual drying of permanent marshes, increased sediment compaction in riverbeds, and destabilization of aquatic habitats. Submerged and floating species including *Vallisneria spiralis* and *Potamogeton* spp. have experienced population declines as they require permanent oxygenated clear waters [23,32]. Studies have reported that the hydrologic regime of the Tigris-Euphrates basin in Iraq is changing, with reduced river discharge, lower groundwater recharge, and more frequent droughts, all of which increase pressure on freshwater ecosystems [41]. Changes to the flow regime, whether caused directly by climate change (e.g., altered rainfall patterns, increased evapotranspiration) or indirectly through water management, can affect aquatic plant morphology by influencing growth conditions such as water depth, current velocity, and light availability. These environmental shifts can lead to alterations in stem elongation, leaf size, root structure, and overall biomass allocation, thereby affecting the success of macrophyte establishment [42,43]. Experimental studies have further demonstrated that increased flow velocity, especially when combined with elevated CO_2_ and dissolved organic carbon concentrations, significantly impacts growth, morphology, and nutrient stoichiometry of submerged macrophytes such as *Berula erecta* [44,45].

### 4.2. Pollution and Water Quality Degradation

Increased pollution from agricultural runoff (mostly pesticides, fertilizers), untreated municipal wastewater, and industrial discharges (primarily from oil industries in southern Iraq) has greatly degraded water quality [46,47,48]. In the Mesopotamian Marshes and the lower Tigris–Euphrates basin, the most common forms of pollution include nutrient enrichment from fertilizers and pesticides, oil-related discharges, and untreated urban wastewater. Several studies [47,48,49,50] have documented high levels of nutrients, heavy metals, and hydrocarbons in marsh waters and sediments, highlighting serious water quality degradation that directly affects native aquatic plants.

Increased nutrient loads are responsible for eutrophication that causes algal blooms and decreases water transparency that is essential for submerged macrophytes [51,52]. In addition, chemical contamination by heavy metals and hydrocarbons inhibits plant growth and can cause direct toxic effects on aquatic plants.

Recent studies have documented the accumulation of heavy metals in several submerged aquatic plant species, including *Vallisneria spiralis*, *Ceratophyllum demersum*, *Potamogeton crispus*, and *Potamogeton perfoliatus*. These findings indicate increasing environmental contamination and suggest that these native macrophytes may be experiencing physiological stress as a result [53].

Such factors disrupt native macrophyte habitats and associated invertebrate communities, as documented by [54,55,56]. Floating invasive species such as *Eichhornia crassipes* increase light limitation for submerged macrophytes while simultaneously inducing another pressure (eutrophication) that affects native the plants [57]. Increased turbidity imposes light limitation, in turn, restricting photosynthesis and growth of submerged macrophytes [58].

### 4.3. Intrusion of Saline Waters

Saltwater intrusion in southern Iraq has intensified significantly in recent decades, primarily due to reduced freshwater discharge from the Euphrates and Tigris Rivers. Upstream damming in Turkey, Syria, and Iran, combined with water diversions within Iraq, has diminished river flow volumes and weakened the hydraulic pressure needed to repel saline water from the Persian Gulf. For example, the Euphrates River’s flow at the Syrian-Iraqi border dropped from around 920 m^3^/s in the 1970s to as low as 197 m^3^/s in recent years. Additionally, Iraq’s practice of diverting saline water from Al Tharthar Lake—where salinity reaches ~1500 ppm—into the Euphrates has raised total dissolved solids in parts of the river to over 1300 ppm [23].

As a result, seawater has advanced deeper inland. In the Shatt al-Arab River, the saltwater intrusion front extended between 83.7 and 112.4 km inland by 2017—substantially farther than in the 1970s, when higher freshwater discharge kept saline waters near the coast [59].

Native aquatic plants such as *Najas marina* and *Trapa natans*, which are poorly adapted to high salinity, have disappeared from many affected areas. Salinity stress impairs plant physiology and reproduction, leading to a shift in community composition toward salt-tolerant species [5,31]. Reviews also note that salinity, alongside other stressors like temperature and nutrient load, negatively affects the abundance and distribution of macrophytes in Iraq [60].

Furthermore, climate change is compounding the issue through reduced rainfall and higher evaporation, which further concentrates salts in the remaining water bodies [41]. Over time, this has degraded structural diversity and habitat complexity, essential for aquatic fauna [23]. Combined with upstream damming and decreased river flow, saltwater intrusion accelerates ecosystem degradation and threatens the resilience of native aquatic vegetation.

### 4.4. Wetland Drainage

UNEP assessments in the early 2000s confirmed the near-total collapse of marsh vegetation following large-scale drainage operations, with catastrophic losses of submerged and emergent aquatic plants [61]. Prolonged sediment exposure to air triggered chemical alterations, including salinization and sulfide accumulation, creating hostile conditions for recolonization [62]. Additionally, disturbance-tolerant species such as *Phragmites australis* expanded rapidly in degraded areas, forming dense stands that reduce open water and diverse habitats needed by submerged and floating plants [23]. The resulting simplification of wetland structure further reduced ecological niches critical for maintaining biodiversity [24].

Substantial reductions in marshland extent and vegetation cover were documented between 1986 and 2000, driven by anthropogenic drainage operations and upstream damming [28,63].

While Huwaiza Marsh exhibited relatively high restoration (approximately 97% of its former vegetative cover), the Chebaish and East Hammar Marshes showed lower recovery levels (61% and 63%, respectively), likely due to salinity intrusion and ongoing habitat degradation. These values represent the estimated proportion of wetland vegetation restored relative to conditions observed in the 1980s prior to large-scale drainage [22].

Similarly, recent ecological surveys of Al-Huwaizah Marsh reported recovery levels of 83% for aquatic plant communities, with submerged, floating, and wetland plant groups exhibiting variable recovery patterns due to salinity stress, water shortages, and pollution from transboundary inflows [33]. Further evidence from the Saffia Nature Reserve, located south of Huwaiza Marsh, indicates that severe reductions in water inflows and increasing salinity have led to the near-total disappearance of submerged aquatic plant species, with only *Phragmites australis* and *Typha domingensis* persisting under extreme conditions [34].

### 4.5. Climate Change

Projected climate change scenarios for the study region indicate rising air and water temperatures, declining annual precipitation, and more frequent and prolonged drought periods. Recent assessments suggest that Iraq’s average temperature could rise by 2.8 °C to over 3 °C by the end of the century, depending on emissions scenarios, while rainfall in northern regions may decrease by 11–21%, further intensifying water scarcity and ecological stress [64]. These changes are expected to exacerbate freshwater shortages, reduce wetland recharge, and challenge the resilience of aquatic ecosystems. These climatic changes are expected to intensify already critical hydrological stress across the region, where water scarcity is a longstanding issue [65,66]. Reduced water discharge will likely diminish the volume and flow variability of freshwater systems such as the Tigris, Euphrates, and Shatt al-Arab [67,68,69,70].

As precipitation declines and temperatures rises, evaporation rates are projected to increase substantially, especially during the summer months. This will not only affect water balance, but also will increase salinity in residual water bodies, particularly detrimental in shallow rivers, marshes, and wetlands [71,72,73]. Increased salinity can disrupt osmotic regulation in aquatic plants, reduce species richness, and favor salt-tolerant or invasive species over native flora. In addition, the higher concentration of nutrients and polluted waters with reduced dilution capacity when evaporation rates are high may lead to eutrophication, algal blooms, and diminished water quality [74,75,76,77].

In this context, submerged aquatic vegetation, already stressed by anthropogenic pollution and hydrologic alteration, will face even greater physiological and ecological pressures under changing climate and salinity regimes [78,79,80]. Recent studies by [81,82] highlight the role of hydrogen peroxide accumulation as a marker of abiotic stress in macrophytes, providing insights into the oxidative stress responses of aquatic species like under environmental fluctuations. In this context, a combined stress by nitrate and heatwaves easily leads to declined photosynthetic efficiency and antioxidant responses [83]. Shifts in temperature and salinity regimes will also affect germination success, photosynthesis, reproduction, and interspecific competition [84,85].

### 4.6. Spread of Invasive Alien and Expanding Native Species

Changing marsh structure due to disturbance driven encroachment of native species such as *Phragmites australis* into monocultures have made certain species of aquatic plants far more challenging to maintain. *Phragmites australis* is native to the region and abundant in some areas that are under highly disturbed hydrological and salinity regimes, which reduces available habitat heterogeneity and reduces potential for submerged and floating plants to persist. Modeling studies suggest *Phragmites australis* spreads aggressively beyond degraded wetlands and into reflooded marsh systems suppressing native aquatic plant species [86]. In monodominant stands of *Phragmites australis*, habitat complexity are reduced [24]. Once established, *Phragmites australis* produces hydrology and shifts to soil chemistry leading to feedback loops that affect native recovery [24]. Although predictive models indicate less suitable habitat under future climate scenarios [87], *Phragmites australis* currently spreads in many degraded wetlands of the study region [86].

Other expanding native species, such as *Arundo donax* and invasive alien species such as *Eichhornia crassipes*, have also spread recently [88]. The spread of these species is facilitated by wetland degradation caused by increased salinity and decreased freshwater flows, especially in the southern marshes [88]. The lack of effective management of invasive alien species at a national level making restoration efforts more difficult [88].

The proliferation of further invasive aquatic plant species such as *Azolla filiculoides* and *Hydrilla verticillata* has been recorded in important wetland areas including the Dalmaj Protected Area [35]. *Azolla* species create dense floating mats which block light and deprive oxygen, while *Hydrilla verticillata* establish dense submerged stands which outcompete native macrophytes [89,90]. Invaded communities are often affected by canal-fed changes to hydrology, and agriculture runoff [35].

Aside from individual threats, recent experimental research on freshwater macrophytes shows that numerous climate change-related stressors (e.g., increased CO_2_, dissolved organic carbon, flow velocities, and eutrophication) can interact with each other in ways that negatively affect plant growth, morphology and nutrient stoichiometry [91]. Interactions that typically have opposing effects, which may hinder predictions and reduce the effectiveness of management interventions applied to aquatic plant communities. Such complexity emphasizes the immediate need for integrated management approaches to conserve the aquatic flora in the Tigris-Euphrates basin.

## 5. Conservation Efforts and Needs

Conservation efforts targeting aquatic ecosystems in the Tigris-Euphrates basin have intensified since the early 2000s, especially following the political changes in Iraq and increasing international recognition of the environmental value of the Mesopotamian Marshes [27]. However, initiatives specifically addressing the protection and restoration of native aquatic plant communities remain limited and fragmented. Wetland restoration initiatives in Iraq have focused heavily on hydrological recovery, while efforts targeting submerged and floating aquatic plants remain scarce. Experiences from other regions highlight the importance of hydrophyte-centered restoration approaches, reducing stressors such as herbivory, eutrophication, and rising salinity [92]. While efforts at marshland restoration have been undertaken after 2003, long-term monitoring indicates persistent challenges related to salinity, sedimentation, and incomplete vegetation recovery [93].

### 5.1. Marshland Restoration Projects

One of the most significant conservation actions was the large-scale re-flooding of the Mesopotamian Marshes starting in 2003. Supported by national efforts and international organizations such as UNEP, UNESCO, and Wetlands International, these projects aimed to restore hydrological connectivity and biodiversity in the drained marshlands [13,27,29]. Restoration activities successfully reintroduced open water habitats and enabled the partial natural recovery of some emergent vegetation, notably *Phragmites australis* and *Typha domingensis* [13,32].

However, restoration outcomes have been highly variable. While some marsh areas, particularly the Al-Hawizeh Marsh, have seen partial recovery of aquatic vegetation, other areas remain degraded due to insufficient water supply, persistent salinity, and pollution [27,94]. Critically, submerged and floating aquatic plants such as *Vallisneria spiralis*, *Najas marina*, and *Potamogeton* species have shown limited or patchy recolonization, reflecting ongoing ecological stress and the absence of targeted restoration strategies [27,32].

### 5.2. International Recognition

The listing of the Ahwar of Southern Iraq (which includes parts of the Mesopotamian Marshes) as a UNESCO World Heritage Site in 2016 brought global attention to the ecological and cultural significance of these wetlands. Additionally, portions of the marshes have been designated as Ramsar Sites, committing Iraq to the international conservation of wetland biodiversity. While these recognitions create a valuable framework for conservation, practical implementation often focuses more heavily on charismatic fauna (e.g., birds, fish) rather than on aquatic plant communities, which receive comparatively little direct management attention [29,95,96,97].

### 5.3. Research and Monitoring Initiatives

Some ecological monitoring programs have been initiated by Iraqi universities, governmental bodies, and NGOs to track biodiversity trends in the marshes and river systems. However, systematic and long-term monitoring of freshwaters remains scarce. Most existing vegetation surveys are either outdated, spatially limited, or lack focus on submerged and floating flora, creating major knowledge gaps regarding the current status of native aquatic plants across the basin Studies such as [31] provide valuable insights into post-restoration aquatic plant dynamics, although comprehensive and long-term basin-wide monitoring programs are still lacking.

### 5.4. Gaps in Current Conservation Strategies

Despite ongoing restoration efforts, several critical gaps remain in current conservation strategies for aquatic plants in the Tigris–Euphrates basin. Notably, there is an absence of species-specific recovery plans targeting threatened aquatic macrophytes, which limits the effectiveness of conservation interventions. Furthermore, Iraq lacks seed bank collections and propagation programs dedicated to aquatic plants, both of which are essential for ex-situ conservation and habitat restoration.

Aquatic vegetation is also insufficiently integrated into broader wetland management and restoration frameworks in Iraq, where attention tends to focus more on charismatic fauna such as birds and fish. Unlike in the European Union for instance where aquatic macrophytes are routinely included in water quality monitoring programs alongside physicochemical parameters (e.g., under the EU Water Framework Directive), Iraq currently lacks a systematic, nationwide monitoring program that incorporates macrophyte diversity and abundance as biological indicators. Although some academic studies have assessed aquatic vegetation in marshes such as Al-Huwaizah and East Hammar [22,33], these efforts remain site-specific and are not integrated into long-term environmental monitoring. Lastly, community engagement in aquatic plant conservation remains minimal, despite the significant cultural and economic importance of wetland vegetation for local livelihoods.

## 6. Recommendations for the Conservation of Native Aquatic Plants

Restoration strategies should incorporate hydrophyte revegetation techniques, including seed-based and transplant-based approaches, supported by prior mesocosm testing and species-specific habitat matching, to maximize restoration success under the challenging environmental conditions of the Tigris-Euphrates basin [92].

Conservation and restoration initiatives need to consider the phenotypic plasticity of aquatic plants; adaptive plastic responses to changes in salinity, drought, and temperature are critical for the persistence of species under progressing climate change [98]. While the rapid pace of climate change will make conservation strategies increasingly challenging, adaptive plasticity, and ecological resilience for conservation priority species will be paramount, as plants that adapt rapidly will be less conservation reliant [99,100]. Restoration initiatives should employ adaptive management strategies that allow for a flexible response in the face of unforeseen environmental changes, this would be in line with developing best practices in climate uncertainty for plant conservation [101].

Conservation and restoration strategies must account for the cumulative and interactive effects of multiple environmental stressors, as aquatic macrophytes often respond to salinity, nutrient enrichment, flow alteration, and thermal stress in combination rather than in isolation. Experimental studies have demonstrated these complex interactions, including the synergistic effects of eutrophication and elevated temperatures [44,45]. To effectively address these complex interactions, it is critical to implement trait-based approaches and adaptive management strategies that enhance ecosystem resilience under projected climate change scenarios [91].

To address the threats to native aquatic plants of the Tigris-Euphrates basin a multi-level conservation approach will be needed. Based on the analysis of current and potential conservation challenges and gaps we have identified several management priorities.

### 6.1. Develop and Implement Species-Specific Recovery Plans

Threatened native aquatic plants like *Vallisneria spiralis*, *Najas marina* and *Trapa natans* should have dedicated recovery plans. These should include habitat restoration, propagation methods and reintroduction programs that consider the species’ ecology [102,103,104,105].

### 6.2. Establish Aquatic Plant Seed Banks and Propagation Programs

Due to the precarious conservation status of many aquatic species, establishing seed banks and vegetative propagation facilities is imperative. These programs could help preserve genetic diversity and provide a stepping stone for active restoration within degraded or restored wetlands [103,106,107,108,109].

### 6.3. Integrate Aquatic Plants into Wetland Management Frameworks

Future wetland conservation and management plans (e.g., Ramsar Site Management Plans) must explicitly include the protection and monitoring of aquatic plant communities. Aquatic plants should be recognized as key ecological indicators of wetland health alongside fauna [110,111,112,113].

### 6.4. Enhance Water Quality Management

Efforts to improve water quality such as controlling agricultural runoff, treating urban and industrial wastewater, and reducing salinity intrusion are critical for creating suitable conditions for aquatic plant recovery. Integrated watershed management approaches involving upstream and downstream stakeholders should be prioritized [114,115,116].

In summary, successful conservation of native aquatic macrophytes will depend on addressing foundational hydrological and water quality issues first. Restoration efforts must integrate wastewater treatment, flow regulation, and sediment control as core components, rather than secondary considerations. Without these prerequisites, even well-designed habitat restoration is unlikely to yield long-term ecological success.

### 6.5. Promote Cross-Border Hydrological Cooperation

Since the Tigris–Euphrates basin is shared by multiple countries, effective transboundary water management is essential to safeguard downstream ecosystems. Cooperative agreements should aim to guarantee minimum environmental flows defined as the quantity, timing, and quality of water required to sustain the ecological health of the marshes and river systems. Maintaining such flows is critical for preserving aquatic plant communities and preventing further habitat degradation in the southern wetlands of Iraq [39,117,118,119,120].

### 6.6. Conduct Regular Monitoring and Research

A coordinated long-term monitoring program is needed to track aquatic plant distributions, population trends, and ecological responses to restoration activities. Research should also focus on understanding the ecological requirements and stress tolerances of native aquatic species [121,122,123,124].

### 6.7. Raise Public Awareness and Engage Local Communities

Marsh vegetation had always been used by local communities for sustenance, and, thus, they are already facilitators in conservation. Further conserving marsh habitats and promoting their value in the local community, by raising the awareness of the ecological role of aquatic plants and by involving local communities in the restoration and monitoring of plants, will increase conservation success [125,126,127].

## 7. Conclusions

The Tigris–Euphrates basin is home to large wetlands that harbor a remarkable assemblage of native aquatic plants, which play a vital role in sustaining biodiversity, water quality, and the livelihoods of local communities. However, decades of hydrological alteration, pollution, salinization, habitat loss, and climate change have severely reduced populations of submerged, floating, and emergent macrophytes. Despite the global significance of the Mesopotamian Marshes, conservation actions have traditionally prioritized more visible fauna, leaving aquatic plants underrepresented in wetland management and policy frameworks.

To reverse this trend, urgent and coordinated conservation actions are needed. In addition to the targeted recovery plans and habitat restoration discussed in this review, it is essential to establish a national aquatic plant monitoring program to track population trends, detect early declines, and guide adaptive management. Developing local capacity through training and supporting researchers and practitioners will ensure that expertise is available to implement effective restoration and management interventions.

Transboundary hydrological cooperation among riparian countries must be strengthened to secure minimum environmental flows, which are critical to sustaining aquatic vegetation and the overall health of the basin’s wetlands. Implementing integrated watershed management practices upstream will help to reduce pollution loads and salinity intrusion that threaten aquatic plant habitats downstream.

Equally important is involving local communities and stakeholders as active partners in conservation efforts. Community-based stewardship, awareness programs, and the revitalization of traditional practices can help to protect and sustainably use wetland resources while reinforcing the cultural and economic value of native aquatic flora.

Protecting and restoring native aquatic plants will not only enhance the ecological resilience of the Tigris–Euphrates basin but also safeguard the cultural heritage and livelihoods of the millions of people who depend on these fragile ecosystems. A stronger commitment at national and regional levels, coupled with science-based restoration, policy reforms, and cross-sectoral collaboration, is essential to secure a sustainable future for Iraq’s unique wetland biodiversity.

## Figures and Tables

**Figure 2 plants-14-01914-f002:**
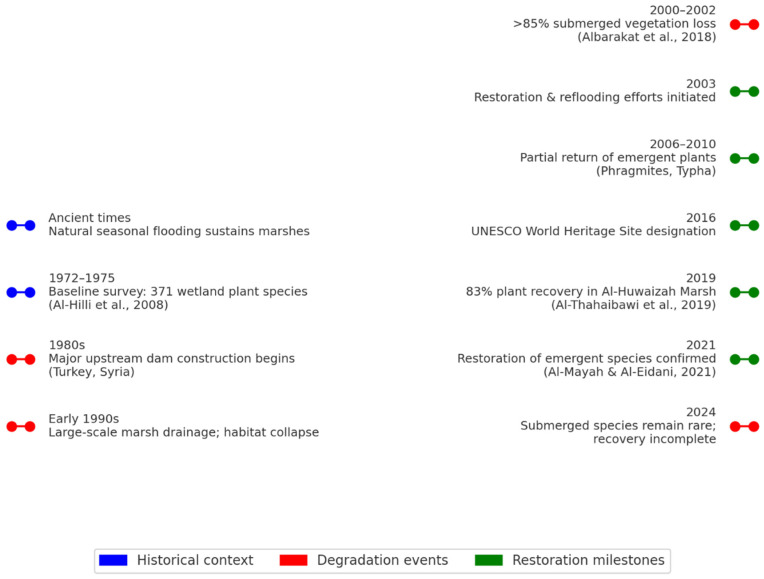
Timeline of major ecological degradation and restoration events affecting the Mesopotamian Marshes in southern Iraq. Red dots indicate degradation phases (e.g., dam construction, drainage, habitat loss), green dots represent restoration or international conservation milestones, and blue dots provide historical context.

**Table 1 plants-14-01914-t001:** Historic and current status of selected aquatic plant species in the Mesopotamian Marshes (Al-Hammar, Al-Huwaiza, and Central Marshes), based on floristic records and ecological assessments.

Species	References	Historic Status	Current Status
*Cyperus papyrus*	[22,32]	Historically scattered	Now critically reduced due to habitat alteration and marsh drainage
*Myriophyllum spicatum*	[22,32]	Historically present in deep or semi-permanent water	Still locally present but declining; sensitive to turbidity and organic pollution
*Najas marina*	[32,33]	Historically recorded in shallow submerged zones	Now rare or possibly extirpated; affected by increased salinity and habitat desiccation
*Potamogeton crispus*	[22,32,33]	Historically widespread	Declining; attributed to pollution, turbidity, and potential heavy metal accumulation
*Stuckenia pectinata*	[32,33]	Historically present in low-salinity marshes	Still persists in some degraded systems; considered tolerant to eutrophication and moderate pollution, but may decline under extreme salinity or desiccation
*Schoenoplectus litoralis*	[22,32,34]	Historically abundant in shallow zones	Now declining; driven by wetland desiccation and increasing salinity
*Trapa natans*	[32,35]	Historically present in freshwater marshes	Now likely extinct or extremely rare; vulnerable to salinity and drainage
*Typha domingensis*	[22,34]	Historically common emergent species	Still present, but with patchy distribution; affected by pollution and marsh fragmentation
*Vallisneria spiralis*	[22,32,33]	Historically dominant submerged macrophyte	Currently rare or absent; impacted by turbidity, flow alteration, and pollution

## Data Availability

Data are contained within the article.

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
