# Peer review of "Threatened Aquatic Plants of the Southern Tigris-Euphrates Basin: Status, Threats, and Conservation Priorities"

_plants, 2025, doi:10.3390/plants14131914_

Round 1
Reviewer 1 Report
Comments and Suggestions for Authors
The manuscript entitled “Threatened Aquatic Plants of the southern Tigris-Euphrates Basin: Status, Threats, and Conservation Priorities” reviewed the distribution, biodiversity, problems, threats and conservation strategies of aquatic vegetation in the Tigris-Euphrates Basin, the origin of ancient Babylon culture. As a researcher on macrophytes, I personally consider that the review is of high value and I learn a lot from this article. The defect of this manuscript is a lack of data and figures. In addition, the writing of this manuscript needs improvement and the phrase is ambiguous more or less. Detailed comments are as follows:
- Line 32, insert a reference herein. Moreover, what is habitat structures? Would you replace this rephrase?
- Line 35-37, a long sentence. Split it.
- Line 48, delete “more well-known”. Try to use fewer adjectives.
- Line 57-58, what are “resilient species”? Which species are resilient species?
- Line 62, southern portion or southern part?
- Line 83, what does “obligate or facultative species”mean?
- Line 91, deeper permanent waters? What is the range of depth? Submerged species will die in deep water since light cannot penetrate.
- Line 104-106, who witnessed the environmental pressures? Try to rephrase the sentence and make it understandable.
- Line 114-115, why the small patches reflect the incomplete recovery? Would you show several photographs?
- Line 169, in the hydrological alterations chapter, would the authors show several data or graphs showing the hydrological alterations?
- Line 197-198, please rephrase the sentence. It is difficult to understand.
- Line 199-200, is likely toxic to aquatic plants.
- Line 204-205, delete “as a result”.
- Line 218, have dropped.
- Line 226, which poorly adapt to.
- Line 244, what is limiting habitat availability? Rephrase.
- Line 275, temperature rises.
- Line 285, delete “by”.
- Line 297, reduce potential?
Author Response
Comments and Suggestions for Authors
The manuscript entitled “Threatened Aquatic Plants of the southern Tigris-Euphrates Basin: Status, Threats, and Conservation Priorities” reviewed the distribution, biodiversity, problems, threats and conservation strategies of aquatic vegetation in the Tigris-Euphrates Basin, the origin of ancient Babylon culture. As a researcher on macrophytes, I personally consider that the review is of high value and I learn a lot from this article. The defect of this manuscript is a lack of data and figures. In addition, the writing of this manuscript needs improvement and the phrase is ambiguous more or less. Detailed comments are as follows:
- Line 32, insert a reference herein. Moreover, what is habitat structures? Would you replace this rephrase?
Response: We thank the reviewer for this valuable suggestion. We have clarified the wording and added the appropriate reference. Specifically, we have replaced the vague term “habitat structures” with “physical habitat complexity” to better convey the ecological function. We also ensured that the existing reference [3] (Thomaz and Cunha, 2010) is directly cited to support this statement.
The revised sentence now reads:
“…nutrient cycling, and enhancing physical habitat complexity for a wide array of organisms [1,2,3].”
- Line 35-37, a long sentence. Split it.
Response:
We appreciate the reviewer’s suggestion. We have revised the long sentence by splitting it into two clearer sentences to improve readability.
- Line 48, delete “more well-known”. Try to use fewer adjectives.
Response:
We thank the reviewer for pointing this out. We have simplified the phrase by removing “more well-known” to reduce redundancy and adjective use.
- Line 57-58, what are “resilient species”? Which species are resilient species?
Response:
We thank the reviewer for this helpful question. In this context, “resilient species” refers to aquatic macrophytes that are widespread and able to tolerate habitat disturbance, salinity, and hydrological fluctuations. A clear example is Phragmites australis, which can persist and expand even under degraded wetland conditions. We have specified this in the sentence to clarify the term.
- Line 62, southern portion or southern part?
Response:
We appreciate the reviewer’s attention to wording. We have revised “southern portion” to “southern part” for simplicity and consistency with common usage.
- Line 83, what does “obligate or facultative species”mean?
Response:
We thank the reviewer for pointing this out. We agree that this phrase could be clarified for readers who may not be familiar with wetland terminology. We have revised the text to explain that “obligate or facultative wetland species” means species that either require wetland conditions to survive (obligate) or can tolerate both wetland and non-wetland conditions (facultative).
- Line 91, deeper permanent waters? What is the range of depth? Submerged species will die in deep water since light cannot penetrate.
Response:
We appreciate the reviewer’s insightful comment. We agree that the phrase “deeper permanent waters” could be misleading without context. In this study area, submerged macrophytes such as Vallisneria spiralis and Potamogeton spp. typically occur in shallow to moderately deep waters, generally ranging from 0.5 to 2 meters, where sufficient light penetration supports photosynthesis year-round. We have revised the wording to specify this depth range and clarify that “permanent” refers to stable water presence rather than great depth.
- Line 104-106, who witnessed the environmental pressures? Try to rephrase the sentence and make it understandable.
Response:
We thank the reviewer for this helpful comment. We recognize that the original wording was ambiguous. We have rephrased the sentence to clearly convey that the Mesopotamian Marshes experienced severe environmental pressures during the 1980s and 1990s due to drainage and altered river flows.
- Line 114-115, why the small patches reflect the incomplete recovery? Would you show several photographs?
Response:
We thank the reviewer for this important comment. Small, isolated patches of submerged macrophytes indicate that environmental conditions (e.g., turbidity, salinity, or hydrological instability) are still suboptimal for the widespread and continuous growth of these species which contrasts with their historically extensive and dense coverage. Thus, patchy distribution serves as an ecological indicator of incomplete recovery.
Regarding photographs, while we agree that visual evidence would strengthen the manuscript, unfortunately, we do not currently have high-resolution, publication-quality field photographs covering all surveyed sites and representative patches. However, we will consider adding photographic documentation in future field surveys to illustrate vegetation recovery status more effectively.
- Line 169, in the hydrological alterations chapter, would the authors show several data or graphs showing the hydrological alterations?
Response:
We thank the reviewer for this valuable suggestion. We agree that visualizing hydrological changes would enhance the understanding of flow reductions and their impacts. However, this review article synthesizes existing literature and does not include original hydrological datasets. We have, instead, incorporated relevant quantitative information in the text (e.g., the reduction of Euphrates River discharge from ~920 m³/s in the 1970s to ~197 m³/s in recent years) and cited the appropriate sources.
To keep the manuscript concise and focused on the aquatic plants, we have not added new figures at this time. However, we fully acknowledge the importance of illustrating hydrological trends and will consider including summary graphs and flow trend data in future expanded studies or supplementary materials.
- Line 197-198, please rephrase the sentence. It is difficult to understand.
Response:
We thank the reviewer for this helpful comment. We have rephrased the sentence to make it clearer and more concise.
- Line 199-200, is likely toxic to aquatic plants.
Response:
We thank the reviewer for this comment. We agree that the phrase should be clearer and more precise. We have revised the wording to explicitly describe the link between heavy metal contamination and toxicity to aquatic plants.
- Line 204-205, delete “as a result”.
Response:
We thank the reviewer for this suggestion. We have removed “as a result” to make the sentence more concise.
- Line 218, have dropped.
Response:
We appreciate the reviewer’s attention to verb tense consistency. We have revised “have dropped” to the simple past tense “dropped” for clarity and grammatical accuracy in this context.
- Line 226, which poorly adapt to.
Response:
We thank the reviewer for highlighting this awkward phrasing. We have revised it for grammatical accuracy and clarity.
- Line 244, what is limiting habitat availability? Rephrase.
Response:
We appreciate the reviewer’s comment and agree that the original phrasing could be clearer. In this context, we intended to convey that the rapid spread of Phragmites australis in degraded areas forms dense stands, which reduces the open water and diverse microhabitats needed by submerged and floating plants. We have rephrased the sentence for clarity.
- Line 275, temperature rises.
Response:
We thank the reviewer for pointing this out. We agree that this phrase should be adjusted for clarity and grammatical correctness.
- Line 285, delete “by”.
Response:
We thank the reviewer for noting this. We have deleted the unnecessary “by” to improve sentence flow.
- Line 297, reduce potential?
Response:
We thank the reviewer for pointing this out. We agree the phrase was unclear and have rephrased it for clarity and precision.
Reviewer 2 Report
Comments and Suggestions for Authors
A well-written paper. My largest complaint is that the conclusions could expanded with more recommendations, say, a 50% increase.
Minor errors are the percentages could be rounded up, e.g., Al-Hammar (86.78%), Central (98.73%), and Al-Huwaiza (83.71%) should read Al-Hammar (87%), Central (99%), and Al-Huwaiza (84%), or to at least one decimal point.
Line 66: 'Tigris and Euphrates Rivers' should read 'Tigris and Euphrates rivers'.
Author Response
Comments and Suggestions for Authors
A well-written paper. My largest complaint is that the conclusions could expanded with more recommendations, say, a 50% increase.
Minor errors are the percentages could be rounded up, e.g., Al-Hammar (86.78%), Central (98.73%), and Al-Huwaiza (83.71%) should read Al-Hammar (87%), Central (99%), and Al-Huwaiza (84%), or to at least one decimal point.
Response:
We sincerely thank the reviewer for this encouraging feedback and constructive suggestions.
Conclusions: We have expanded the Conclusion section by approximately 50% to include additional clear and actionable recommendations. This extension emphasizes practical steps such as the establishment of a national aquatic plant monitoring program, capacity-building for local researchers, strengthening transboundary water governance, and promoting community-based stewardship to enhance conservation success.
Percentages: We have rounded the reported marsh vegetation loss percentages for clarity and consistency, following the reviewer’s guidance:
Al-Hammar: from 86.78% → 87%
Central: from 98.73% → 99%
Al-Huwaiza: from 83.71% → 84%
Line 66: 'Tigris and Euphrates Rivers' should read 'Tigris and Euphrates rivers'.
Response:
We thank the reviewer for noting this capitalization inconsistency. We have corrected ‘Tigris and Euphrates Rivers’ to ‘Tigris and Euphrates rivers’ to follow standard capitalization for common nouns.
Reviewer 3 Report
Comments and Suggestions for Authors
General comments
The review is well documented with a logical sequence of ideas. Please, find below some suggestion I`ve proposed in order to improve the manuscript quality.
Specific comments
Line 196. The phrase “The Mesopotamian Marshes, consist of several wetland complexes, i.e., the Al-Ham-96 mar, Al-Hawizeh, and the Central Marshes ” is repeated.
Line 107. What were the restauration efforts and what were the results that you mention? The mentioned species multiplied? Or? Please, include the year of the survey for a better timeline understanding. Try to describe the events in a logical timeline.
Line 133. Describe what sort of human activities interfere with the aquatic plant communities.
Line 194. Please, state what type of pollution is more common in the study area and if there are some studies regarding the pollution assessment conducted in the study area.
Line 214. There are studies that conclude that statement? If there is, please, make reference.
Line 229. Please use the same typology of cite the references through all the manuscript.
Line 320. Please, rephrase with a subject and a predicate.
Line 366. Take time in order to correct the grammar and the punctuation signs.
Author Response
Comments and Suggestions for Authors
General comments
The review is well documented with a logical sequence of ideas. Please, find below some suggestion I`ve proposed in order to improve the manuscript quality.
Specific comments
Line 196. The phrase “The Mesopotamian Marshes, consist of several wetland complexes, i.e., the Al-Ham-96 mar, Al-Hawizeh, and the Central Marshes ” is repeated.
Response:
We thank the reviewer for catching this unintentional repetition. We have carefully reviewed the section and removed the redundant sentence to avoid duplication and improve readability.
Line 107. What were the restauration efforts and what were the results that you mention? The mentioned species multiplied? Or? Please, include the year of the survey for a better timeline understanding. Try to describe the events in a logical timeline.
Response:
Following the large-scale drainage of the Mesopotamian Marshes during the 1990s, significant restoration efforts began after 2003 with extensive re-flooding supported by national initiatives and international partners such as UNEP and UNESCO. Post-restoration ecological surveys conducted between 2006 and 2008 (e.g., the Habitat Mapping and Monitoring Project) and subsequent studies up to 2017 documented partial recovery of emergent macrophytes such as Phragmites australis and Typha domingensis in marshes like Al-Hammar and Al-Huwaiza. However, sensitive submerged species such as Vallisneria spiralis, Najas marina, and Potamogeton spp. remained rare or confined to small patches, indicating incomplete recovery due to persistent salinity, pollution, and altered hydrology.
Line 133. Describe what sort of human activities interfere with the aquatic plant communities.
Response:
Major human activities impacting aquatic plant communities in the Mesopotamian Marshes include large-scale wetland drainage for land reclamation, upstream dam construction and water diversions reducing river flow, expansion of oil extraction infrastructure, discharge of untreated municipal and industrial wastewater, intensive agriculture leading to fertilizer and pesticide runoff, and overharvesting of wetland vegetation for local fuel and fodder.”
Line 194. Please, state what type of pollution is more common in the study area and if there are some studies regarding the pollution assessment conducted in the study area.
Response:
We thank the reviewer for this valuable suggestion. We have clarified the types of pollution that are most prevalent in the Mesopotamian Marshes and the lower Tigris–Euphrates basin, specifically noting nutrient enrichment from agriculture, oil industry discharges, and untreated municipal wastewater. We have also added references to recent studies that assess water quality and document the presence of nutrients, heavy metals, and hydrocarbons.
Line 214. There are studies that conclude that statement? If there is, please, make reference.
Response:
We thank the reviewer for this valuable comment. We have now added appropriate references to support the statement regarding the incomplete recovery of aquatic plant communities and the continued impact of human activities.
Line 229. Please use the same typology of cite the references through all the manuscript.
Response:
We thank the reviewer for highlighting this issue. We have carefully checked the manuscript and standardized the citation style throughout to ensure consistency.
Line 320. Please, rephrase with a subject and a predicate.
Response:
We thank the reviewer for highlighting this issue. We have revised the sentence at Line 320 to ensure it is a complete sentence with a clear subject and predicate. The revised version now reads:
“Projected climate change scenarios for the study region indicate rising air and water temperatures, declining annual precipitation, and more frequent and prolonged drought periods.”
Line 366. Take time in order to correct the grammar and the punctuation signs.
Response:
We appreciate the reviewer’s attention to language accuracy. We have carefully revised the sentence at Line 366 to correct grammar and punctuation and to ensure clarity
Round 2
Reviewer 1 Report
Comments and Suggestions for Authors
Since the present work describes the natural history of macrophytes in the Tigris-Euphrates River Basin and does not involve with experimental design and statistical analyses, I do not and can not have too many problems with it.